# ACTIVE AUTOMATED MACHINE LEARNING WITH SELF-TRAINING

## ABSTRACT

Automated Machine Learning (AutoML) aims to automatically select and configure machine learning algorithms for optimal performance on given datasets. In real-world applications, training data oftentimes contain a large amount of unlabeled examples, whereas the amount of labeled examples is limited. However, AutoML tools have so far only focused on supervised learning, i.e., utilizing labeled data for training, leaving the valuable information provided by unlabeled data untapped. To address this limitation, we introduce our augmented AutoML system AutoActiveSelf-Labeling (AutoASL), which combines principles from self-training and active learning to effectively leverage unlabeled data during the training process. AutoASL iteratively self-labels previously unlabeled data instances, which is achieved through a powerful ensemble of AutoML and traditional ML algorithms, resulting in a substantial expansion of the labeled training data. We observe synergetic effects between the incorporated self-training and active learning components, leading to an improvement of the overall accuracy compared to state-of-the-art tools.

## 1 INTRODUCTION

In the rapidly evolving landscape of machine learning, AutoML has emerged with the promise to democratize the application of machine learning by automating the complex process of selecting proper learning algorithms and optimizing their hyperparameters. State-of-the-art tools have shown impressive results on a variety of tasks, especially for tabular data (Thornton et al., 2013; Feurer et al., 2015; Mohr et al., 2018; Olson & Moore, 2019; Hollmann et al., 2023). However, these tools are all operating within the confines of supervised learning, relying heavily on the quality and amount of labeled data for model training and evaluation.

In practice, datasets do not always conform to the supervised setting. Labeled data is oftentimes limited, while a vast amount of unlabeled data remains untapped. Acquiring labeled data can be time-consuming and expensive, e.g., medical data such as electronic health records (EHRs), where experts are required to provide annotations (Mugisha & Paik, 2023). However, AutoML has largely ignored this crucial aspect so far.

In this context, *semi-supervised learning* (SSL) becomes relevant. Aiming to leverage both labeled and unlabeled data to enhance model performance (Chapelle et al., 2006), SSL encompasses various techniques such as consistency regularization, entropy minimization, and pseudo-labeling, which are widely employed to harness the latent information within unlabeled data (Wallin et al., 2022; Zhao et al., 2022). These methods make use of one or more underlying assumptions of the data distribution, including smoothness, cluster structure, or manifold properties, to guide the learning process (Ouali et al., 2020). For instance, Darabi et al. (2021) and Yoon et al. (2020) propose different data augmentation strategies for tabular data and employ consistency regularization. The idea is to encourage the model to produce consistent predictions across these different, augmented views of the same data instance, which substantially improves the robustness of the learner. In contrast, Varma & Ré (2018) adopt a pseudo-labeling approach. They propose the tool *Snuba*, which automates the process of labeling unlabeled data. *Snuba* selects base learners such as decision trees or logistic regressors, based on their diversity and accuracy on the labeled training dataset. These base learners then predict pseudo-labels for the unlabeled instances, which are combined.

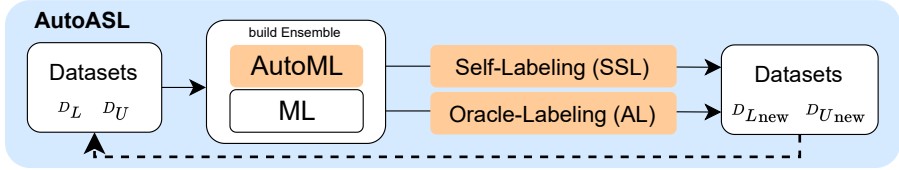

Figure 1: Our proposed approach AutoActiveSelf-Labeling (AutoASL). We build ensembles of AutoML and traditional ML algorithms and incorporate a self-labeling and an active learning component to update the labeled dataset $\mathcal{D}_L$ and the unlabeled dataset $\mathcal{D}_U$.

Closely related to SSL is the idea to further optimize the utilization of labeled data through *active learning* (AL). In AL (Dasgupta et al., 2008; Beygelzimer et al., 2010), the goal is to strategically select the most informative instances from an unlabeled pool for manual annotation (Nguyen et al., 2019; Gao et al., 2020). This process can be carried out in a variety of ways, including batch selection and interactive modes, where an oracle (typically a human expert) is requested to provide labels for specific instances.

While SSL and AL are closely related, they have mostly been studied in isolation. However, Zhu et al. (2003) combine them under a Gaussian random field and minimize its energy function, whereas Gao et al. (2020) employ consistency regularization. Both approaches additionally integrate AL by providing pseudo-supervision from an oracle for specific, unlabeled instances.

However, the potential synergy between AutoML, SSL, and AL has not been explored so far. In this paper, our primary goal is to harness the principles and methodologies of SSL and AL jointly to integrate them within the context of AutoML (see Figure 1). Thereby, we enable AutoML systems to be applied to semi-supervised data. More concretely, our main contributions include the following:

1. *AutoActiveSelf-Labeling (AutoASL).* We introduce a novel and efficient algorithm, designed specifically for semi-supervised tabular data tasks. AutoASL combines traditional ML and AutoML algorithms and incorporates strategies from SSL and AL to leverage information from unlabeled data.

2. *AutoML for SSL* (Section 3). We explore the synergies between AutoML, SSL and AL and demonstrate how methods from AutoML can effectively address SSL tasks.

3. *Application.* We efficiently implement AutoASL, evaluate it on a rich set of diverse datasets and compare it to existing methods. The implementation can be found in the supplementary material [1] and will be made open source upon acceptance.

## 2 PROBLEM DEFINITION AND NOTATION

For the sake of simplicity, we begin with the binary classification scenario, where we are given a $d$-dimensional feature space $\mathcal{X} \in \mathbb{R}^d$. Each instance $\mathbf{x}_i = (x_i^1, ..., x_i^d) \in \mathcal{X}$ is (non-deterministically) associated with a label $y_i \in \mathcal{Y} = \{0, 1\}$ via a joint probability distribution $\mathbb{P}$. A dataset is then given as a sample $\mathcal{D} = \{(\mathbf{x}_i, y_i)\}_{i=1}^n$ from this joint probability distribution. In this setting, the goal is to find a hypothesis $h : \mathcal{X} \to \mathcal{Y}$ from a hypothesis space $\mathcal{H}$ that minimizes the generalization error (risk) with respect to a given loss function $\ell : \mathcal{Y} \times \mathcal{Y} \to \mathbb{R}^+$:

$$h^* \in \arg\min_{h \in \mathcal{H}} \int_{(\mathbf{x}_i, y_i) \sim \mathbb{P}} \ell(h(\mathbf{x}_i), y_i) \, d\mathbb{P}$$

The latter is approximated by the empirical risk minimizer, e.g., by dividing $\mathcal{D}$ into a training set $\mathcal{D}_{\text{train}}$ and test set $\mathcal{D}_{\text{test}}$, and finding

$$\hat{h} = \arg\min_{h \in \mathcal{H}} \frac{1}{|\mathcal{D}_{\text{test}}|} \sum_{(\mathbf{x}_i, y_i) \in \mathcal{D}_{\text{test}}} \ell(h(\mathbf{x}_i), y_i) \ ,$$

where $\mathcal{D}_{\text{train}}$ is used to induce the hypothesis $h$.

---

[1]https://anonymous.4open.science/r/AutoASL-3E44/

One way of finding this optimal classifier is to apply AutoML. Given a set of learning algorithms $\mathcal{A} = \{A^{(1)}, \ldots A^{(k)}\}$ with corresponding hyperparameter spaces $\Lambda^{(1)}, \ldots, \Lambda^{(k)}$, a learning algorithm induces a hypothesis given a dataset from a dataset space $\mathbb{D}$ and a hyperparameter setting $A^{(j)} : \mathbb{D} \times \Lambda^{(j)} \rightarrow \mathcal{H}$. AutoML seeks to find the most suitable algorithm and hyperparameter setting $A^*_{\lambda^*}$:

$$A^*_{\lambda^*} \in \underset{A^{(j)} \in \mathcal{A}, \lambda \in \Lambda^{(j)}}{\operatorname{argmin}} \frac{1}{|\mathcal{D}_{\text{test}}|} \sum_{(\mathbf{x}_i, y_i) \in \mathcal{D}_{\text{test}}} \ell(h(\mathbf{x}_i), y_i), \text{ where } h := A^{(j)}(\mathcal{D}_{\text{train}}, \lambda).$$

AutoML has proven to deliver fast and accurate solutions for the supervised setting (Erickson et al., 2020; Mohr & Wever, 2021; Hollmann et al., 2023). However, current AutoML tools assume full supervision, whereas in the setting of SSL, not every instance $\mathbf{x}_i \in \mathcal{D}_{\text{train}}$ is associated with a label $y_i \in \mathcal{Y}$. Instead, we are given a small labeled dataset $\mathcal{D}_L = \{(\mathbf{x}_i, y_i)\}_{i=1}^{l}$ and a large unlabeled dataset $\mathcal{D}_U = \{(\mathbf{x}_i)\}_{i=l+1}^{n}$ (i.e., $|\mathcal{D}_L| \ll |\mathcal{D}_U|$). In this setting, several options are conceivable for an AutoML tool:

- *Reduction to supervised learning (SL):* One straightforward approach is to simply ignore the unlabeled data $\mathcal{D}_U$ and solely use $\mathcal{D}_L$ for model induction. This has the advantage that standard supervised learning techniques suffice, but the obvious disadvantage of wasting an opportunity to improve performance through additional training information.

- *Semi-supervised learning (SSL):* An arguably better approach that avoids this disadvantage is to apply SSL methods to the entire data $\mathcal{D}_L \cup \mathcal{D}_U$. This, of course, requires an extension of standard AutoML tools, so as to enable them to handle unlabeled data. In our approach, to be detailed in the next section, this will be accomplished by means of self-training, where the learner is first trained on $\mathcal{D}_L$ and then assigns pseudo-labels to selected instances from $\mathcal{D}_U$, thereby increasing its (labeled) training data.

- *Active learning (AL):* Instead of self-labeling additional training examples, labels could also be queried from an oracle, for example, a human expert. Compared to self-labeling, feedback by the oracle will presumably be more reliable. On the other side, human experts are slow, costly, and do not scale to large amounts of (unlabeled) data.

In addition to opting for any of these "pure" strategies, the AutoML tool may of course also apply them in a combined way, i.e., querying the oracle for a certain part of $\mathcal{D}_U$, self-labeling another part, and perhaps ignoring the rest. Thereby, an optimal compromise between benefit and cost could be achieved. Indeed, while SL is less costly than SSL, which in turn is less costly than AL, the order is exactly reversed in terms of (expected) benefit: The presumably most useful (reliable) information is provided by AL, followed by SSL, and finally SL. The design of an optimal "mixed" strategy, tailored to the context and concrete problem at hand, can be seen as an interesting learning or reasoning task on a *meta-level* (Hüllermeier et al., 2021).

## 3 AUTOMATED ACTIVE SELF-LABELING

We propose the approach AutoASL, which combines supervised learning, SSL, and AL. Throughout this paper we utilize the same taxonomy as introduced by Triguero et al. (2015). We first outline the core functionality of AutoASL. Following an iterative procedure, we initially train an ensemble of learners on the labeled dataset $\mathcal{D}_L$. Subsequently, the learners generate pseudo-labels for the instances within the unlabeled dataset $\mathcal{D}_U$. Based on the consensus or disagreement among the learners for each instance, we either (a) self-label the instance, (b) forward it to an oracle for labeling, (c) identify the instance as challenging to label or (d) leave it inside $\mathcal{D}_U$.

We then update the datasets $\mathcal{D}_L$ and $\mathcal{D}_U$ as follows. Instances that have been either self-labeled or labeled by the oracle are incorporated into the training dataset $\mathcal{D}_L$, while instances identified as difficult to label are excluded from $\mathcal{D}_U$. This marks the initiation of the next iteration, where the ensemble is retrained on the newly adjusted $\mathcal{D}_L$, and the process continues iteratively.

### 3.1 SELF-TRAINING

As already said, self-training (Triguero et al., 2015) is a strategy in which the learner, previously trained on $\mathcal{D}_L$, pseudo-labels instances contained in $\mathcal{D}_U$. Thus, an expanded training dataset $\mathcal{D}_{L_{self}}$

---

**Algorithm 1** AutoActiveSelf-Labeling (the detailed pseudocode is given in Appendix 2)

---

**Require:** Labeled dataset $\mathcal{D}_L$, Unlabeled dataset $\mathcal{D}_U$, Maximum iterations $n$, Set of learning algorithms $\mathcal{A}$, MetaEnsemble $\mathcal{E} := [\text{TabPFN}]$, Oracle $O$, Thresholds $\tau, \sigma, \rho$, Number of self-labeled instances $s$, Number of oracle-labeled instances $o$, Initial iteration $iter = 0$

1:  $\tilde{\mathcal{A}} \leftarrow \text{SELECT}(\mathcal{D}, \mathcal{A}, \tau)$          ▷ *Select classifiers with 5-fold CV-score $> \tau$ on $\mathcal{D}_L$.*
2:  $c \leftarrow |\tilde{\mathcal{A}}|$
3:  $\mathcal{E} \leftarrow \mathcal{E} \cup \tilde{\mathcal{A}}$                                ▷ *Construct MetaEnsemble.*
4:  $r \leftarrow \text{COMPUTECLASSPROPORTION}(\mathcal{D}_L)$       ▷ *Computes class proportion of positive class.*
5:  **if** $\tilde{\mathcal{A}} = \emptyset$ **then return** $\mathcal{D}_L$          ▷ *No classifier was selected $\rightarrow$ abstain from self-labeling.*
6:  **end if**
7:  **while** $iter < n$ **do**:
8:     $\{h_i | 1 \leq i \leq c\} \leftarrow \text{TRAIN}(\mathcal{E}, \mathcal{D}_L)$          ▷ *Train MetaEnsemble on $\mathcal{D}_L$.*
9:     $\hat{Y} \leftarrow h_i(\mathcal{D}_U)$                   ▷ *Predict hard labels for $\mathcal{D}_U$ for all $h_i$.*
10:    $\mathbf{z}_1 \leftarrow p_1(\mathcal{D}_U)$                   ▷ *Predict class probabilities for $\mathcal{D}_U$.*
11:    $\mathcal{D}_{L_{self}}^{conf}, \mathcal{D}_U^{unconf}, \mathcal{D}_U^{rest} \leftarrow \text{ASSIGNSETS}(\mathcal{D}_U, \hat{Y}, \mathbf{z}_1, \sigma, \rho, c)$
12:    $DS \leftarrow \mathcal{D}_U^{rest}$
13:    $AS, OS \leftarrow \text{CONSTRUCTSETS}(\mathcal{D}_{L_{self}}^{conf}, \mathcal{D}_U^{unconf}, s, o, r)$
14:    **if** $AS = \emptyset$ **then return** $\mathcal{D}_L$    ▷ *MetaEnsemble $\mathcal{E}$ is uncertain $\rightarrow$ abstain from self-labeling.*
15:    **end if**
16:    $\mathcal{D}_L \leftarrow \mathcal{D}_L \cup AS \cup OS$                 ▷ *Update labeled dataset $\mathcal{D}_L$.*
17:    $\mathcal{D}_U \leftarrow \mathcal{D}_U \setminus \{DS \cup AS \cup OS\}$       ▷ *Update unlabeled dataset $\mathcal{D}_U$.*
18:    $iter \leftarrow iter + 1$
19: **end while**
20: **return** $\mathcal{D}_L$

---

can be generated, consisting of the original labeled data $\mathcal{D}_L$ and the new pseudo-labeled instances $\{(\mathbf{x}_i, \hat{y}_i)\}_{i=k}^j$, with $\mathbf{x}_i \in \mathcal{D}_U$ and $\hat{y}_i$ representing its pseudo-label. The underlying idea behind self-training is that correctly labeling a portion of the new pseudo-labeled instances leads to an improvement in test accuracy. While self-training is a promising approach, it comes with inherent risks and challenges that must be acknowledged. A primary concern is the potential incorporation of inaccurately labeled instances, leading to what is commonly referred to as self-confirmation bias (Arazo et al., 2020). When the model makes incorrect predictions during pseudo-labeling, these errors can propagate through subsequent training iterations. Therefore, the quality of the pseudo-labeling process is critical to the overall success of self-training (Lienen & Hüllermeier, 2021; Lang et al., 2022; Lienen et al., 2023). Another concern revolves around the possible alteration of the data distribution. Self-training has the capacity to modify the data distribution, e.g., if only instances from one class receive pseudo-labels. This alteration can disrupt the balance between classes, potentially resulting in biased models. As such, it is necessary to actively monitor and mitigate any shifts in the data distribution that may occur during the self-training process. Self-training incurs at a relatively modest cost, yet its effectiveness depends on numerous factors, as discussed above.

### 3.2 MULTI-LEARNING, MULTI-CLASSIFIER METHOD

To build the ensemble, AutoASL first has to select suitable algorithms. Therefore, we consider the set consisting of two different, fast AutoML tools, namely TabPFN (Hollmann et al., 2023) and AutoGluon (Erickson et al., 2020) and ten different simple scikit-learn models (Pedregosa et al., 2011), among them a random forest and a decision tree. For a complete list of the used ML models, we refer to Appendix C. TabPFN is always incorporated in the ensemble, due to its extremely fast runtime (Hollmann et al., 2023), accurate performance, and well-calibrated probabilities, that AutoASL uses later on. Further, AutoASL selects all algorithms from the different AutoML-tools and the simple ML models, that achieve a 3-fold cross-validation score $> \tau$ on $\mathcal{D}_L$ (Algorithm 1, line 1). The parameter $\tau$ has to be set beforehand by the user. We refer to this set of selected algorithms and the TabPFN classifier as MetaEnsemble (Algorithm 1, line 3), since, e.g., AutoGluon on its own constructs an ensemble. In cases where the number of selected algorithms (TabPFN excluded) is even, we remove the worst-performing algorithm. This step is essential to maintain an even count of algorithms in MetaEnsemble, a prerequisite for our uncertainty criterion in this section.

Let in the following $h_1(\mathbf{x}_i)$ be the predicted label for the instance $\mathbf{x}_i$ by TabPFN and $p_1(\mathbf{x}_i)$ its predicted probabilistic label. Let further $h_2(\mathbf{x}_i), ..., h_c(\mathbf{x}_i)$ be the predicted label by the other algorithms inside the MetaEnsemble, with $c$ being the number of selected algorithms ($c = 4$ in Figure 2). Then AutoASL computes the class proportion or ratio $r$ of the positive class of $\mathcal{D}_L$, which will be used in the sampling strategy later on, where AutoASL samples in a stratified manner. Once selected, MetaEnsemble stays fixed for the follow-up iterations. The intuition behind including the simple ML models is to integrate a counterweight consisting of simple, robust, but also diverse models that counteract the potential overfitting and the resulting self-confirmation bias of the AutoML tools. The MetaEnsemble is now trained on $\mathcal{D}_L$ (Algorithm 1, line 9) and then predicts pseudo-labels for each instance in $\mathcal{D}_U$. Hence, each instance receives different predicted pseudo-labels from the different algorithms inside the MetaEnsemble.

### 3.3 CONFIDENCE MEASURES

To mitigate the risk of wrongly labeling instances inside $\mathcal{D}_U$, the following hybrid confidence prediction approach (Triguero et al., 2015) is proposed (Algorithm 1, lines 10-12). AutoASL combines the agreement of the classifiers in the MetaEnsemble with the predicted probabilities of TabPFN. In the first step, only the instances where the predicted labels of all predictors coincide are taken. From those, AutoASL further filters out only the most confident ones. Hereby, AutoASL relies on the well-calibrated probabilistic predictions of the TabPFN predictor while taking only instances into account, where the probabilistic prediction was confident enough, i.e., either $p_1(\mathbf{x}_i) > \sigma$ or $p_1(\mathbf{x}_i) < 1 - \sigma$. We refer to these instances as confident instances $\mathcal{D}_U^{conf}$ and the set of confident instances together with their predicted label as $\mathcal{D}_{L_{self}}^{conf}$ in the following, or formally:

$$\mathcal{D}_U^{conf} = \left\{ \mathbf{x}_i \in \mathcal{D}_U \mid h_1(\mathbf{x}_i) = \ldots = h_c(\mathbf{x}_i), \, p_1(\mathbf{x}_i) \notin [1 - \sigma, \sigma] \right\}$$

$$\mathcal{D}_{L_{self}}^{conf} = \left\{ (\mathbf{x}_i, h_1(\mathbf{x}_i)) \mid x_i \in \mathcal{D}_U^{conf} \right\}$$

### 3.4 SELF-LABELING SAMPLING STRATEGY

We propose not to add all confident instances with their label to $\mathcal{D}_L$, but instead to sample a subset, which we call agreement set ($AS$). To mitigate shifts in the data distribution of the training data, we first employ a stratified sampling approach, which ensures that the class proportion $r$ of the positive class of $AS$ approximately matches that of the initial training dataset $\mathcal{D}_L$. Further, we sample uniformly, such that the instances within $AS$ exhibit varying degrees of dissimilarity, dependent on the chosen confidence threshold $\sigma$.

An alternative sampling approach could involve selecting only the $s$ instances, where the probabilistic prediction of TabPFN is the most confident. However, while this may lead to higher label accuracy compared to our sampling strategy, it may not necessarily improve the generalization performance of the ensemble. The reason for this lies in the likelihood that instances with highly confident predictions tend to be very similar and cluster closely together. Consequently, they may not provide the learner with new information about the underlying data distribution. This issue is also discussed by Zhang & Sabuncu (2020), who emphasize that selecting perhaps less confident yet more diverse instances leads to an increased generalization performance of the learner.

### 3.5 ACTIVE LEARNING COMPONENT

Instances characterized by the highest degree of uncertainty among predictors, i.e., the instances where the number of predictors inside the ensemble that predict label 0 coincides with the number that predicts label 1 (note, that $c$ is even), and additionally, the TabPFN probabilistic prediction falls between $1 - \rho$ and $\rho$, are referred to as $\mathcal{D}_U^{unconf}$, or formally:

$$\mathcal{D}_U^{unconf} = \left\{ \mathbf{x}_i \in \mathcal{D}_U \mid \sum_{j=1}^{c} h_j(\mathbf{x}_i) = \frac{c}{2}, \, p_1(\mathbf{x}_i) \in (1 - \rho, \rho) \right\}$$

We expect the instances within this set to have the greatest potential for information gain, assuming we have knowledge of their true labels (Freund et al., 1997). As we want to present these instances

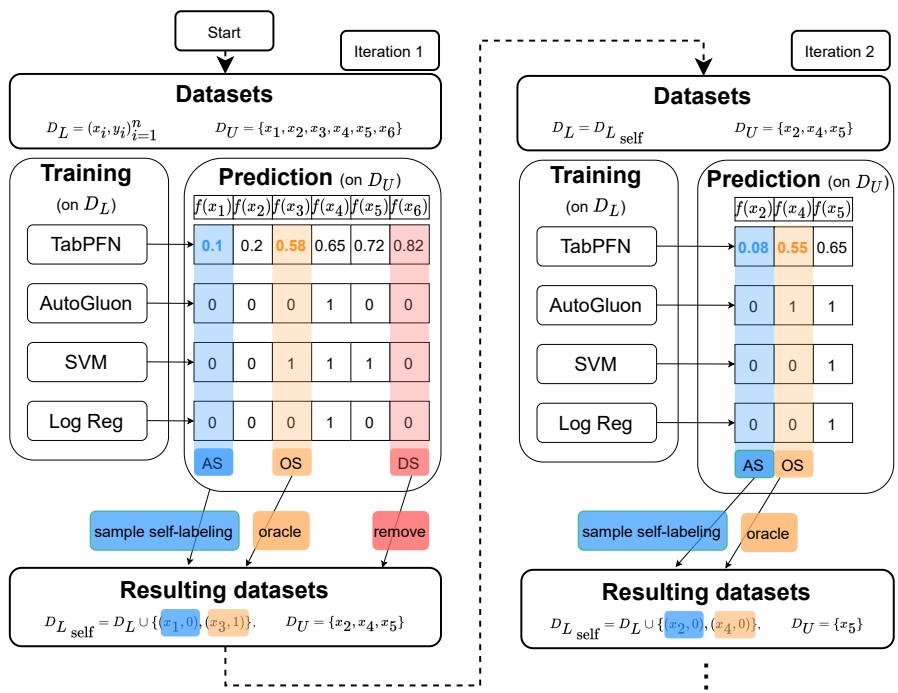

Figure 2: Exemplary explanation of our approach. The selected algorithms are trained on $\mathcal{D}_L$ and predict on $\mathcal{D}_U$. TabPFN predicts probabilities (refer to these as $p_1(\mathbf{x}_i)$ and to the converted hard labels as $h_1(\mathbf{x}_i)$) and the other algorithms in the ensemble predict hard labels (refer to these as $h_2(\mathbf{x}_i), h_3(\mathbf{x}_i), h_4(\mathbf{x}_i)$). If all algorithms agree on a label for a specific instance $\mathbf{x}_i$, e.g., $h_1(\mathbf{x}_i) = h_2(\mathbf{x}_i) = h_3(\mathbf{x}_i) = h_4(\mathbf{x}_i) = 0$ and the TabPFN prediction is confident enough, i.e., $p_1(\mathbf{x}_i) > \sigma$ or $p_1(\mathbf{x}_i) < 1 - \sigma$, in the figure $\sigma = 0.8$, bold blue, the instance will be added to $\mathcal{D}_L$ with its predicted label. If the algorithms are maximally uncertain about a specific instance, e.g., $h_1(\mathbf{x}_i) = h_3(\mathbf{x}_i) = 1$ and $h_2(\mathbf{x}_i) = h_4(\mathbf{x}_i) = 0$ and TabPFN is unconfident, i.e., $1 - \rho < p_1(x_i) < \rho$, in the figure $\rho = 0.6$, bold orange, the instance will be given to and labeled by an oracle. The instance and its label will then be added to $\mathcal{D}_L$ for the next iteration. If a majority of the algorithms disagrees with a minority on a specific instance, e.g., $h_1(\mathbf{x}_i) = 1$ and $h_2(\mathbf{x}_i) = h_3(\mathbf{x}_i) = h_4(\mathbf{x}_i) = 0$, the instance will be removed from $\mathcal{D}_U$. The rest of the instances will stay in $\mathcal{D}_U$ for the followup iteration.

to an oracle for labeling, which is costly in practice, AutoASL again uniformly samples a subset, namely $o$ instances from $\mathcal{D}_U^{unconf}$. In our implementation, however, the oracle has access to the true labels. The sampled subset of instances together with their labels predicted by the oracle is referred to as oracle set ($OS$) in the following. This integration of an oracle, effectively a "human-in-the-loop" (Mosqueira-Rey et al., 2023), has the chance to enhance user confidence in our AutoML system, since it enables users to inject their expertise into the process precisely where our tool exhibits uncertainty, e.g., by harnessing domain knowledge (Lee et al., 2019).

Further, all instances within $\mathcal{D}_U$, for which a majority of the predictors disagrees with a minority, are referred to as the disagreement set ($DS$), or formally:

$$DS = \mathcal{D}_U \setminus \mathcal{D}_U^{unconf} .$$

All instances within $DS$ will be removed from $\mathcal{D}_U$ ($\mathbf{x}_6$ in Figure 2, iteration 1). This prevents them from getting wrongly labeled in future iterations. The sampled instances from $AS$ and $OS$ with their corresponding labels will be added to $\mathcal{D}_L$. The sampled instances from $AS$ and $OS$ will be also removed from $\mathcal{D}_U$.

### 3.6 INCREMENTAL ADDITION MECHANISM

AutoASL retrains the MetaEnsemble on the updated training dataset $\mathcal{D}_{L\text{self}}$. This whole process is repeated for a fixed number of iterations $n$, which incrementally enlarges the size of $\mathcal{D}_{L\text{self}}$.

### 3.7 STOPPING CRITERIA

AutoASL implements three different stopping criteria: (a) In the case that none of the models from which the Ensemble is constructed achieves a sufficiently high accuracy score on $\mathcal{D}_L$, the self--labeling procedure is stopped and abstains from labeling instances of $\mathcal{D}_U$ (Algorithm 1, line 5). In this case, AutoASL is too uncertain to learn the data distribution and does not want to risk to wrongly pseudo-label instances. (b) If the agreement set $AS$ is empty, AutoASL stops the self-labeling process and abstains from labeling as well (Algorithm 1, line 13). In this case, MetaEnsemble would predict only unconfident probabilistic scores. (c) If the maximum number of iterations $n$ is reached, AutoASL stops as well.. If any of these criteria is fulfilled, AutoASL predicts on the test data. Hereby, we rely on TabPFN as a single classifier, which has been trained on the final $\mathcal{D}_{L_{self}}$ (similar to all other algorithms in MetaEnsemble). The whole pipeline is visualized in Figure 2.

### 3.8 PARAMETERS

Our proposed algorithm involves different parameters that influence various trade-offs.

1. *Number of Iterations* $(n)$: The choice of how many iterations to perform is crucial for the performance of AutoASL. In each iteration, the size of the self-labeled dataset $\mathcal{D}_{L_{self}}$ increases. This can enhance the robustness of the algorithm since it is trained on a larger training dataset. However, this increase in iterations comes at the potential cost of decreased accuracy in pseudo-labels, because the probability of mislabeling instances due to factors like confirmation bias and modified data distribution becomes higher.

2. *Instance Sampling (s and o):* The parameters $s$ and $o$ dictate how many instances to sample for the self-labeling and for the oracle from the unlabeled dataset $\mathcal{D}_U$ in each iteration. While a higher $s$ introduces trade-offs similar to those discussed in the previous item, it is important to select $o$ in a reasonable way; typically, $o$ will be relatively low, especially if experts are involved in the labeling process.

3. *Confidence Threshold* $(\sigma)$: The threshold $\sigma$ determines the confidence level for including instances in the self-labeled dataset. A higher $\sigma$ results in higher accuracy of pseudo-labels. However, this comes at the cost of an increased likelihood of substantial shifts in the data distribution of $\mathcal{D}_{L_{\text{self}}}$. Very confident instances tend to cluster in a small region and exhibit high similarity, offering only limited information gain for the learner into the overall underlying data distribution.

4. *Threshold for Model Selection* $(\tau)$: The threshold $\tau$ influences the extent to which models are permitted to be included in the MetaEnsemble. The higher $\tau$, the more capable the models are to learn the data. But this might also be an indicator of overfitting, which would result in lower generalization capability.

## 4 EXPERIMENTS

We evaluate our proposed approach AutoASL on real-world binary classification tasks and compare it against state-of-the-art (supervised) AutoML-tools and semi-supervised methods for tabular data.

### 4.1 DATASETS

We used all open source datasets from the OpenML-CC18 (Vanschoren et al., 2013) benchmark suite that meet the requirements of TabPFN (Hollmann et al., 2023), i.e., no categorical attributes, at most 10 classes and at most 100 features. We further filter by restricting ourselves to binary classification tasks with not too imbalanced class distributions, i.e., class proportion $0.25 < r < 0.75$, leaving us with a set of 47 different datasets.

### 4.2 BASELINES

We conduct an empirical analysis, comparing our approach against six different methods. We evaluate two state-of-the-art supervised AutoML tools, TabPFN (Hollmann et al., 2023) and AutoGluon (Erickson et al., 2020), trained exclusively on the labeled data $\mathcal{D}_L$, and referred to as TabPFN-SL

and AutoGluon-SL, respectively. Subsequently, we explore a basic self-training approach, where both, TabPFN and AutoGluon, are first trained on $\mathcal{D}_L$. Then, they predict pseudo-labels for all instances in the unlabeled data $\mathcal{D}_U$, and are retrained on the expanded training data $\mathcal{D}_{L\text{self}}$. We refer to these learners as TabPFN-SSL and AutoGluon-SSL, respectively. We further compare against the automated self-labeling tool, Snuba (Varma & Ré, 2018), using the existing open-source code[2] provided by the authors. We implemented a shallow multi-layer perceptron as an end-to-end model capable of handling the probabilistic labels generated by Snuba for instances within $\mathcal{D}_U$. Finally, our comparative analysis includes VIME (Yoon et al., 2020), a semi-supervised learning tool designed specifically for tabular data. We utilized the open-source implementation[3] provided by the authors.

## 4.3 EVALUATION

For each dataset and method, we conducted 20 repetitions, each with unique random seeds and dataset splits. All methods utilized the same dataset split when initialized with a specific seed. For every seed value, we randomly sampled a distinct subset from the entire dataset, comprising 1500 instances. This subset was subsequently divided into two portions: 1000 instances for $D_L$ and $D_U$ and 500 instances for the $\mathcal{D}_{\text{test}}$. This partition was necessitated by the limitations of TabPFN, which can only be trained on a maximum of 1000 instances. This is exactly the case, if all instances within $\mathcal{D}_U$ were pseudo-labeled, and thus $|\mathcal{D}_{L\text{self}}| = 1000$. We investigate a split size of 0.025 for $\mathcal{D}_L$ and $\mathcal{D}_U$, hence each containing 25 and 975 instances, respectively. This small split size was chosen to focus on the use case where only very few instances are available. Although one might argue that this data size is quite limited, we align with the viewpoint presented by Hollmann et al. (2023), that small tabular datasets are most often encountered in real-world applications. We have found the following parameters to generally make AutoASL a robust, well performing algorithm: $n = 10$, $s = 15$, $o = 5$, $\tau = 0.75$, $\sigma = 0.8$, $\rho = 0.6$, and thus used them in the experiments.

## 4.4 RESULTS

In Figure 3, we illustrate the ranking of each optimization method based on its test accuracy using critical distance plots. These rankings are calculated by averaging the results from all datasets and runs. By assigning ranks to performance measures, these plots provide a more intuitive and comprehensive understanding of the relative performance. As can be seen, AutoASL ranks better than any other approach considered across the different runs and datasets. Note that the improvement of AutoASL over the baselines is significant.

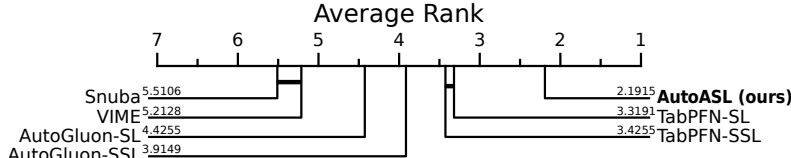

Figure 3: Average rank plot for the performance measure Accuracy.

In Table 1, we present the results from 47 tabular datasets. After conducting the self-labeling process, we computed the average test accuracy across 20 different seeds for each algorithm. It is evident that our approach demonstrates superior performance when compared to other methods. Specifically, AutoASL outperforms all other approaches in 22 of the 47 datasets.

Having incorporated a stopping criterion that hinges on the prediction confidence of TabPFN, our approach becomes dependent on the performance of TabPFN. Hence in cases, where TabPFN performs quite well, our method improves notably (e.g., OpenML IDs 44, 803, and 823).

Overall, AutoASL is able to achieve state-of-the-art performance. Thanks to the combination of techniques from AutoML, SSL, and AL, our proposed approach outperforms existing approaches.

---

[2]https://github.com/HazyResearch/reef
[3]https://github.com/jsyoon0823/VIME

Table 1: Average test accuracy (mean ± standard deviation) measured after the self-labeling process. Best performances for a dataset are highlighted in bold.

| OpenML IDs | 44 | 293 | 351 | 354 | 715 | 722 | 723 | 727 |
|---|---|---|---|---|---|---|---|---|
| AutoGluon | .828 ± .07 | .553 ± .06 | .686 ± .06 | .504 ± .02 | **.594 ± .07** | .701 ± .09 | **.561 ± .05** | .743 ± .08 |
| TabPFN | .892 ± .02 | .575 ± .05 | .854 ± .04 | .506 ± .02 | .571 ± .05 | .785 ± .05 | .550 ± .02 | .761 ± .04 |
| VIME | .595 ± .27 | .497 ± .25 | .210 ± .34 | **.566 ± .37** | .526 ± .18 | .385 ± .28 | .520 ± .19 | .551 ± .16 |
| Snuba | .591 ± .18 | **.587 ± .06** | .813 ± .04 | .491 ± .07 | .517 ± .10 | **.798 ± .06** | .455 ± .10 | .760 ± .06 |
| **AutoASL (Ours)** | **.903 ± .02** | .574 ± .05 | **.878 ± .04** | .508 ± .02 | .569 ± .05 | .794 ± .04 | .549 ± .03 | **.800 ± .04** |

| OpenML IDs | 734 | 735 | 740 | 743 | 751 | 752 | 761 | 772 |
|---|---|---|---|---|---|---|---|---|
| AutoGluon | .703 ± .08 | .835 ± .05 | .600 ± .05 | .643 ± .08 | .606 ± .07 | .526 ± .05 | .841 ± .04 | .521 ± .03 |
| TabPFN | .772 ± .04 | .870 ± .02 | .573 ± .02 | .638 ± .03 | .588 ± .06 | .527 ± .03 | .878 ± .03 | .538 ± .03 |
| VIME | .637 ± .24 | .809 ± .16 | **.746 ± .16** | .613 ± .23 | **.774 ± .16** | .492 ± .16 | .775 ± .19 | .459 ± .29 |
| Snuba | .444 ± .14 | .434 ± .20 | .483 ± .12 | .603 ± .10 | .496 ± .15 | .507 ± .12 | .356 ± .22 | **.543 ± .10** |
| **AutoASL (Ours)** | **.784 ± .04** | **.880 ± .02** | .582 ± .03 | **.653 ± .05** | .601 ± .05 | **.528 ± .03** | **.892 ± .02** | .537 ± .03 |

| OpenML IDs | 797 | 799 | 803 | 806 | 807 | 813 | 816 | 819 |
|---|---|---|---|---|---|---|---|---|
| AutoGluon | .563 ± .06 | .712 ± .05 | .880 ± .08 | **.586 ± .05** | .625 ± .06 | .691 ± .05 | .655 ± .08 | .793 ± .06 |
| TabPFN | .566 ± .02 | .775 ± .04 | .896 ± .03 | .570 ± .03 | **.674 ± .05** | .658 ± .07 | .701 ± .08 | .830 ± .03 |
| VIME | **.634 ± .27** | .393 ± .13 | .304 ± .36 | .539 ± .21 | .523 ± .24 | **.725 ± .16** | .522 ± .17 | .403 ± .21 |
| Snuba | .451 ± .12 | .728 ± .06 | .873 ± .10 | .489 ± .13 | .659 ± .06 | .594 ± .12 | .675 ± .06 | .825 ± .03 |
| **AutoASL (Ours)** | .571 ± .02 | **.794 ± .04** | **.921 ± .02** | .570 ± .03 | .672 ± .05 | .672 ± .08 | **.723 ± .08** | **.839 ± .03** |

| OpenML IDs | 821 | 822 | 823 | 833 | 837 | 843 | 845 | 846 |
|---|---|---|---|---|---|---|---|---|
| AutoGluon | .757 ± .04 | .686 ± .07 | .903 ± .10 | .650 ± .06 | **.571 ± .07** | .780 ± .07 | .605 ± .07 | .681 ± .09 |
| TabPFN | .771 ± .04 | .752 ± .05 | .943 ± .02 | .697 ± .02 | .562 ± .03 | .812 ± .04 | .700 ± .05 | .746 ± .03 |
| VIME | .220 ± .32 | .379 ± .34 | .443 ± .27 | .133 ± .21 | .560 ± .23 | .186 ± .29 | .586 ± .19 | .264 ± .18 |
| Snuba | .568 ± .31 | .635 ± .19 | .418 ± .28 | **.822 ± .03** | .484 ± .11 | .692 ± .22 | .616 ± .10 | **.757 ± .11** |
| **AutoASL (Ours)** | **.777 ± .03** | **.761 ± .04** | **.959 ± .01** | .691 ± .02 | .562 ± .03 | **.819 ± .03** | **.711 ± .06** | .751 ± .03 |

| OpenML IDs | 847 | 849 | 866 | 871 | 901 | 903 | 904 | 910 |
|---|---|---|---|---|---|---|---|---|
| AutoGluon | .790 ± .05 | .622 ± .07 | .606 ± .07 | **.503 ± .03** | .689 ± .06 | .624 ± .07 | .562 ± .07 | .608 ± .06 |
| TabPFN | .804 ± .02 | .629 ± .04 | .597 ± .02 | **.503 ± .02** | .728 ± .05 | .593 ± .03 | .606 ± .03 | .596 ± .04 |
| VIME | .364 ± .31 | .517 ± .20 | **.762 ± .14** | .495 ± .17 | .607 ± .15 | **.655 ± .21** | .408 ± .14 | **.681 ± .21** |
| Snuba | .807 ± .03 | .573 ± .14 | .541 ± .11 | .480 ± .08 | .673 ± .07 | .531 ± .18 | **.634 ± .05** | .515 ± .16 |
| **AutoASL (Ours)** | **.821 ± .01** | **.633 ± .04** | .595 ± .02 | .501 ± .02 | **.742 ± .06** | .601 ± .05 | .606 ± .03 | .607 ± .06 |

| OpenML IDs | 912 | 913 | 917 | 979 | 1120 | 1489 | 1494 | |
|---|---|---|---|---|---|---|---|---|
| AutoGluon | **.765 ± .06** | .674 ± .06 | .558 ± .05 | .724 ± .05 | .691 ± .09 | .762 ± .02 | .710 ± .06 | |
| TabPFN | .710 ± .08 | .629 ± .06 | .567 ± .04 | .753 ± .03 | .720 ± .04 | .741 ± .02 | .748 ± .05 | |
| VIME | .660 ± .24 | **.682 ± .22** | .565 ± .23 | **.862 ± .08** | .668 ± .29 | **.859 ± .09** | .735 ± .15 | |
| Snuba | .709 ± .12 | .616 ± .09 | .468 ± .11 | .458 ± .22 | .390 ± .16 | .377 ± .25 | .407 ± .20 | |
| **AutoASL (Ours)** | .733 ± .10 | .656 ± .07 | **.570 ± .04** | .782 ± .03 | **.728 ± .05** | .747 ± .03 | **.749 ± .05** | |

# 5 CONCLUSION AND FUTURE WORK

We have shown that AutoASL, combining principles and algorithms from AutoML, SSL, and AL, yields superior performance compared to existing state-of-the-art AutoML- and SSL-tools. This result hopefully encourages further exploration and innovation in the AutoML-community to tackle real-world problems falling within the SSL-setting. Indeed, we consider our work merely a first step in this direction, leaving much room for further improvements. For example, we have exclusively focused on the binary classification setting so far, leaving an extension of our framework to the multi-class setting for future work, which should be doable in a more or less straightforward way.

As already mentioned in the end of Section 2, it might also be tempting to tackle our approach from the broader perspective of (optimal) metareasoning, a view of AutoML that has recently been advocated by Hüllermeier et al. (2021): An AutoML tool is an agent that has to train a model on a given set of data, which is the main reasoning task. Finding a good way of doing so requires deliberation on a meta-level, including, e.g., decisions about the ML pipeline to be used, and in our setting also about the labeling and the data to train on. AutoASL can be seen as a simple metareasoning strategy, in which, however, many decisions are still made in an ad-hoc manner (including, for example, the labeling strategy and the hard-coding of parameters outlined in Section 3.8). Building on the theory of optimal metareasoning and bounded rationality (Russell, 1997; Cox & Raja, 2011), better approaches can presumably be developed in a more principled manner. Moreover, by combining metareasoning with meta-learning (Brazdil et al., 2022), tools like AutoASL should be able to automatically improve over the course of time.

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
