Felix Mohr and Marcel Wever. Naive Automated Machine Learning - A Late Baseline for AutoML. *CoRR*, abs/2103.10496, 2021.

Felix Mohr, Marcel Wever, and Eyke Hüllermeier. ML-Plan: Automated machine learning via hierarchical planning. *Mach. Learn.*, 107(8-10):1495–1515, 2018.

Eduardo Mosqueira-Rey, Elena Hernández-Pereira, David Alonso-Ríos, José Bobes-Bascarán, and Ángel Fernández-Leal. Human-in-the-loop machine learning: a state of the art. *Artif. Intell. Rev.*, 56(4):3005–3054, 2023.

Chérubin Mugisha and Incheon Paik. Bridging the gap between medical tabular data and nlp predictive models: A fuzzy-logic-based textualization approach. *Electronics*, 12(8):1848, 2023.

Vu-Linh Nguyen, Sébastien Destercke, and Eyke Hüllermeier. Epistemic uncertainty sampling. In *Discovery Science - 22nd International Conference, DS 2019, Split, Croatia*, volume 11828 of *Lecture Notes in Computer Science*, pp. 72–86. Springer, 2019.

Randal S. Olson and Jason H. Moore. TPOT: A Tree-Based Pipeline Optimization Tool for Automating Machine Learning. In *Automated Machine Learning - Methods, Systems, Challenges*, The Springer Series on Challenges in Machine Learning, pp. 151–160. Springer, 2019.

Yassine Ouali, Céline Hudelot, and Myriam Tami. An overview of deep semi-supervised learning. *CoRR*, abs/2006.05278, 2020.

Fabian Pedregosa, Gaël Varoquaux, Alexandre Gramfort, Vincent Michel, Bertrand Thirion, Olivier Grisel, Mathieu Blondel, Peter Prettenhofer, Ron Weiss, Vincent Dubourg, Jake VanderPlas, Alexandre Passos, David Cournapeau, Matthieu Brucher, Matthieu Perrot, and Edouard Duchesnay. Scikit-learn: Machine Learning in Python. *J. Mach. Learn. Res.*, 12:2825–2830, 2011.

S. Russell. Rationality and intelligence. *Artificial Intelligence*, 94:57–77, 1997.

Chris Thornton, Frank Hutter, Holger H. Hoos, and Kevin Leyton-Brown. Auto-WEKA: combined selection and hyperparameter optimization of classification algorithms. In *The 19th ACM SIGKDD International Conference on Knowledge Discovery and Data Mining, KDD 2013, Chicago, IL, USA, August 11-14, 2013*, pp. 847–855. ACM, 2013.

Isaac Triguero, Salvador García, and Francisco Herrera. Self-labeled techniques for semi-supervised learning: taxonomy, software and empirical study. *Knowl. Inf. Syst.*, 42(2):245–284, 2015.

Laurens van der Maaten and Geoffrey Hinton. Visualizing data using t-SNE. *Journal of Machine Learning Research*, 9:2579–2605, November 2008.

Joaquin Vanschoren, Jan N. van Rijn, Bernd Bischl, and Luís Torgo. OpenML: networked science in machine learning. *SIGKDD Explor.*, 15(2):49–60, 2013.

Paroma Varma and Christopher Ré. Snuba: Automating Weak Supervision to Label Training Data. *Proc. VLDB Endow.*, 12(3):223–236, 2018.

Erik Wallin, Lennart Svensson, Fredrik Kahl, and Lars Hammarstrand. Doublematch: Improving semi-supervised learning with self-supervision. In *26th International Conference on Pattern Recognition, ICPR 2022, Montreal, QC, Canada*, pp. 2871–2877. IEEE, 2022.

Jinsung Yoon, Yao Zhang, James Jordon, and Mihaela van der Schaar. VIME: Extending the Success of Self- and Semi-supervised Learning to Tabular Domain. In *Advances in Neural Information Processing Systems 33: Annual Conference on Neural Information Processing Systems*, 2020.

Zhilu Zhang and Mert R. Sabuncu. Self-distillation as instance-specific label smoothing. 2020.

Zhen Zhao, Luping Zhou, Lei Wang, Yinghuan Shi, and Yang Gao. Lassl: Label-guided self-training for semi-supervised learning. pp. 9208–9216. AAAI Press, 2022.

Xiaojin Zhu, Zoubin Ghahramani, and John Lafferty. Combining Active Learning and Semi-Supervised Learning Using Gaussian Fields and Harmonic Functions. In *Proceedings of the 20th International Conference on Machine Learning (ICML)*, pp. 912–919, 2003.

# A ANALYSIS OF THE PSEUDO-LABELING STRATEGY

In the following we visually analyze our proposed pseudo-labeling strategy. In Figure 4 the dataset with id 44 and seed 0 was selected. Further a split of 0.025 was chosen and AutoASL constructed the MetaEnsemble out of TabPFN, Gaussian Naive Bayes, AutoGluon and Random Forest.

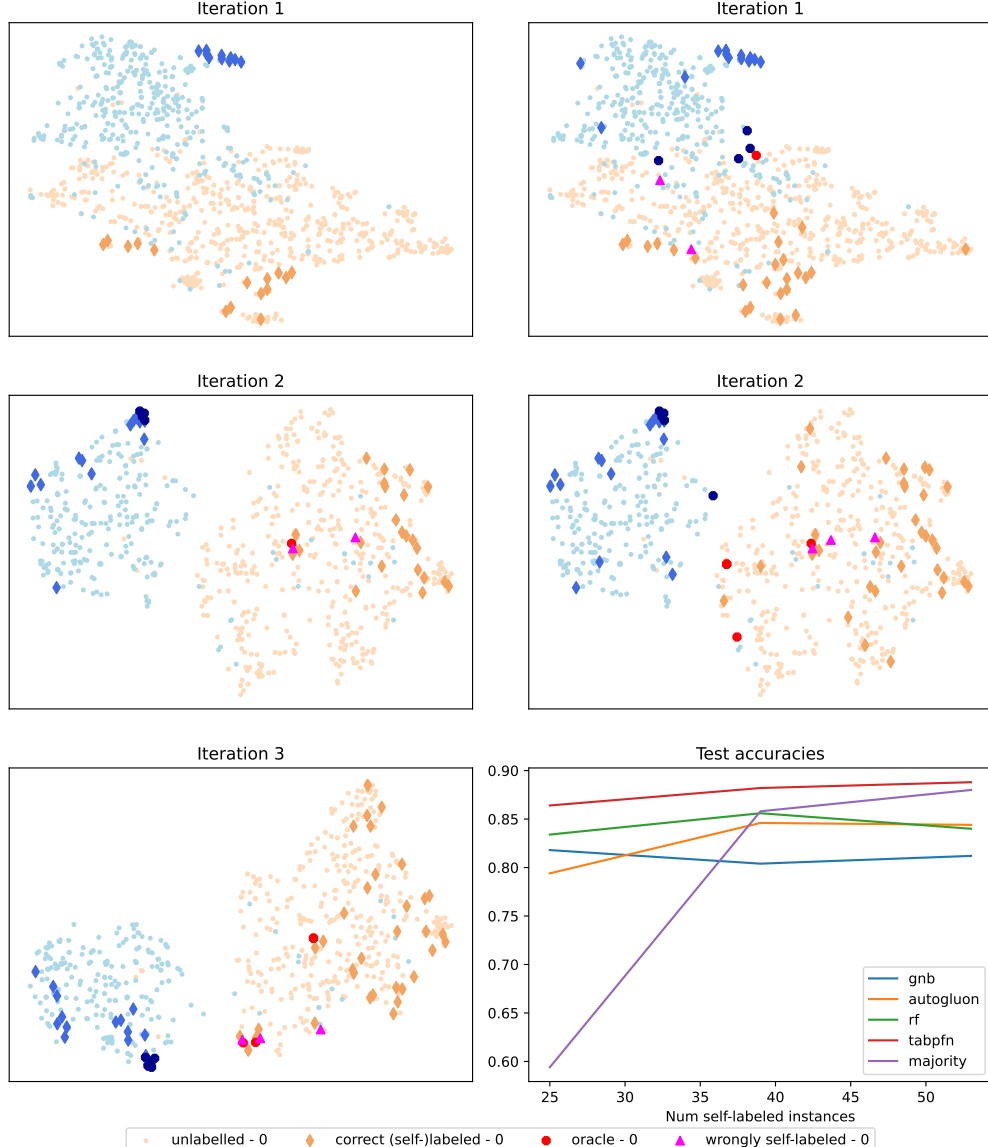

Figure 4: TabPFN embeddings for the data instances during the first iterations.

We present visualizations of the TabPFN embeddings of the data instances from $\mathcal{D}_L$ (diamonds in iteration 1, top left subfigure) and $\mathcal{D}_U$ (small circles in iteration 1, top left subfigure), after projecting them down to a 2-dimensional subspace using t-sne van der Maaten & Hinton (2008). For this specific run, we made the following parameter choices: $\tau = 0.75$, $\sigma = 0.8$, $\rho = 0.6$, $s = 15$, $l = 5$, and $n = 5$. In the first iteration TabPFN was trained on the initial labeled dataset. Then, we generated the embeddings by passing all instances within $D_L$ and $D_U$ through the encoder of

TabPFN. As can be clearly seen, TabPFN is able to separate the instances from both classes quite well. The dataset $\mathcal{D}_L$ is expanded by incorporating pseudo-labeled instances, which received labels from both, AutoASL and the oracle. As can be seen in the subfigure on the top right, the self-labeled instances exhibit a broad distribution across the instance space, aligning with our objective of uniformly sampling diverse instances (the instances that are now visualized as diamonds). The instances labeled by the oracle (red and dark blue) are situated in regions where instances belonging to both labels are located, i.e., close to the decision boundary, which reflects the uncertainty of AutoASL in these regions. Consequently, these instances were selected for being labeled by the oracle. All pseudo-labeled instances are subsequently integrated into $\mathcal{D}_L$. The subsequent subfigure (iteration 2, middle left) showcases the embeddings after having trained TabPFN on the expanded $\mathcal{D}_L$, before again generating the TabPFN embeddings. This iterative process continues. Despite some mislabeled instances the test accuracy consistently increases, as shown in the final subfigure.

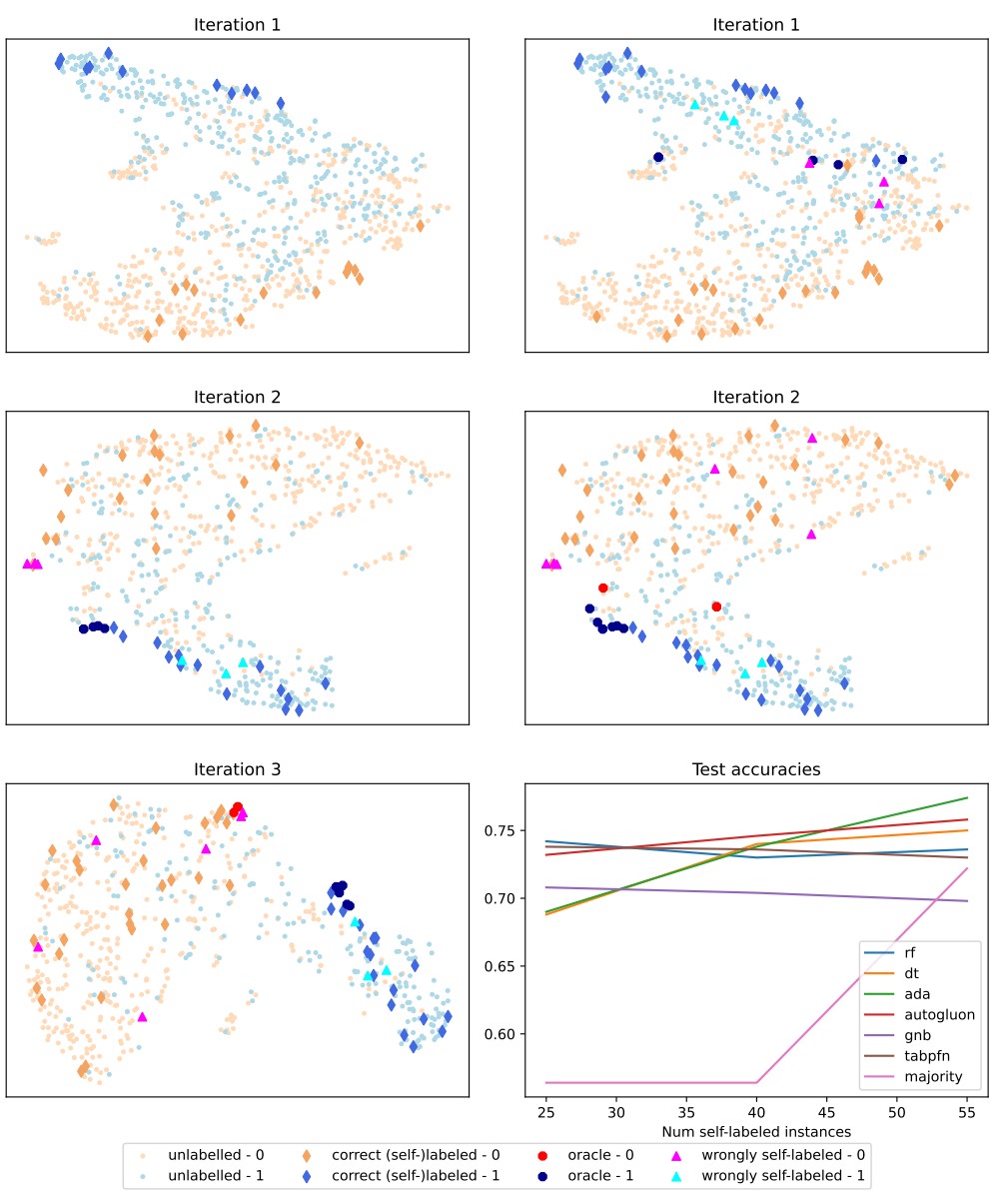

Figure 5: TabPFN embeddings for the data instances during the first iterations.

In Figure 5 the dataset with id 734, seed 1 was selected. Further, a split of 0.025 was chosen and AutoASL constructed the MetaEnsemble out of TabPFN, Random Forest, Ada Boost, Gaussian Naive Bayes, AutoGluon, and Support Vector Machine. In Figure 6 we analyzed the pseudo-labeling strategy for the dataset with id 351, seed 3, split 0.025.

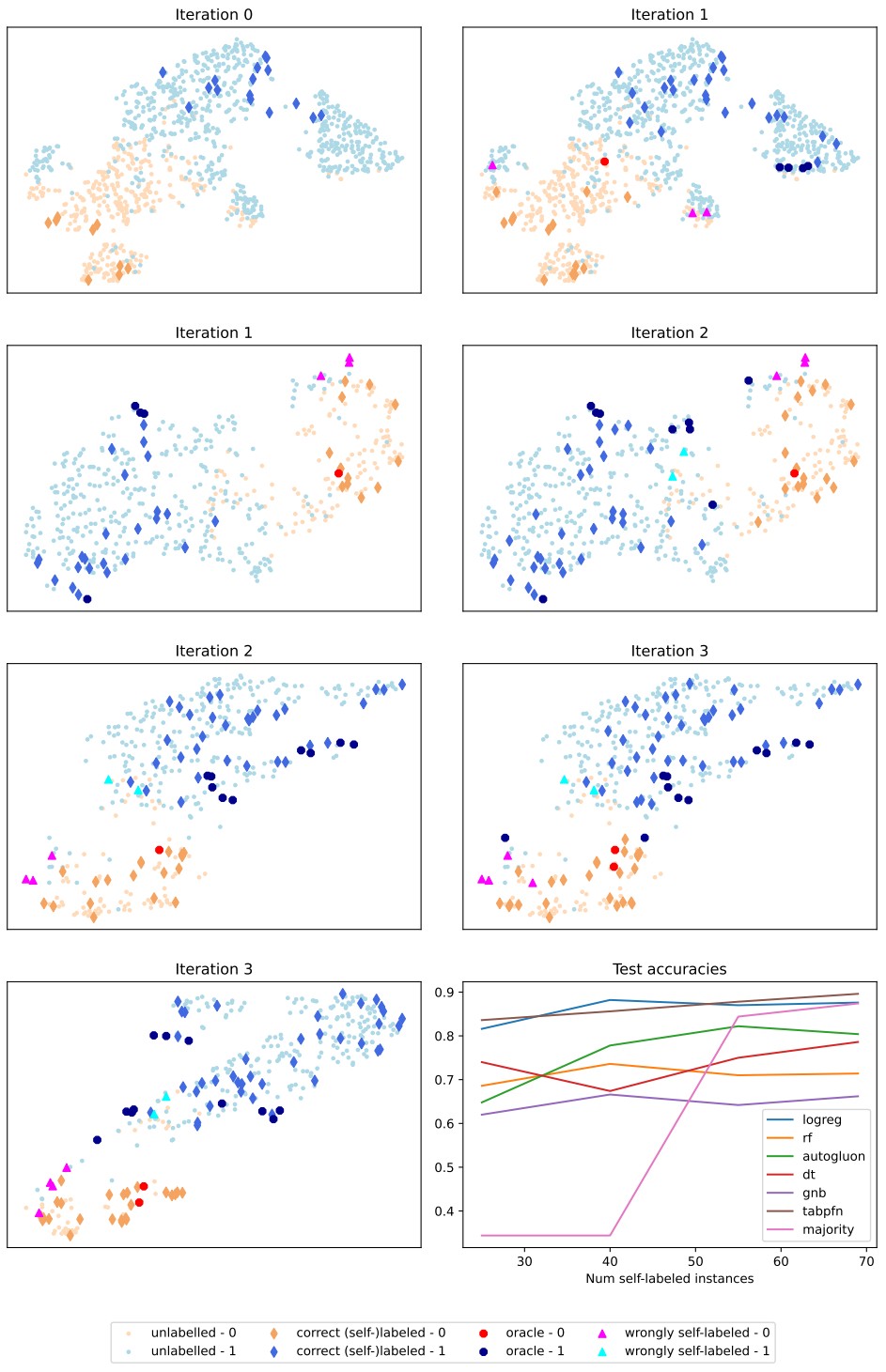

Figure 6: TabPFN embeddings for the data instances during the first iterations.

## B    IMPLEMENTATION

In the following we describe the detailed pseudo-code for AutoASL. In Algorithm 2 we again describe the overall structure of AutoASL. First, a number of suitable algorithms is selected based on their performance on the labeled dataset $\mathcal{D}_L$ to construct the MetaEnsemble. Then, we train the selected algorithms on the dataset $\mathcal{D}_L$ and predict hard labels as well as class probabilities for the instances in $\mathcal{D}_U$. Based on these predictions and different thresholds, that had to be chosen beforehand, Algorithm 3 splits $\mathcal{D}_U$ into different sets, namely a confident, self-labeled dataset $\mathcal{D}_{L_{self}}^{conf}$, an uncertain, unlabeled dataset $\mathcal{D}_U^{unconf}$ and an unlabeled dataset $\mathcal{D}_U^{rest}$, that consists of all remaining instances within $\mathcal{D}_U$. From these datasets, Algorithm 4 constructs the oracle set $OS$, the disagreement set $DS$ and the agreement set $AS$. Then, the labeled and the unlabeled dataset $\mathcal{D}_L$ and $\mathcal{D}_U$, respectively, are updated for the next iteration.

---

**Algorithm 2** AutoActiveSelf-Labeling

---

**Require:** Labeled dataset $\mathcal{D}_L$, Unlabeled dataset $\mathcal{D}_U$, Maximum iterations $n$, Set of learning systems $\mathcal{A}$, MetaEnsemble $\mathcal{E}$ {TabPFN}, Oracle $O$, Thresholds $\tau, \sigma, \rho$, Number of self-labeled instances $s$, Number of oracle-labeled instances $o$, Initial iteration $iter = 0$

  1: $\tilde{\mathcal{A}} \leftarrow \text{SELECT}(\mathcal{D}_L, \mathcal{A}, \tau)$              ▷ *Select classifiers with 5-fold CV-score $> \tau$ on $D_L$.*
  2: $c \leftarrow |\tilde{\mathcal{A}}|$
  3: $\mathcal{E} \leftarrow \mathcal{E} \cup \tilde{\mathcal{A}}$                                ▷ *Construct MetaEnsemble.*
  4: $p \leftarrow \text{COMPUTECLASSPROPORTION}(\mathcal{D}_L)$      ▷ *Computes class proportion of positive class.*
  5: **if** $\tilde{\mathcal{A}} = \emptyset$ **then return** $\mathcal{D}_L$      ▷ *No classifier was selected $\rightarrow$ abstain from self-labeling.*
  6: **end if**
  7: **while** $iter < n$ **do**:
  8:      $\{h_i | 1 \leq i \leq c\} \leftarrow \text{TRAIN}(\mathcal{E}, \mathcal{D}_L)$               ▷ *Train MetaEnsemble on $\mathcal{D}_L$.*
  9:      $\hat{Y} \leftarrow h_i(\mathcal{D}_U)$                    ▷ *Predict hard labels for $\mathcal{D}_U$ for all $h_i$.*
10:      $\mathbf{z} \leftarrow p_1(\mathcal{D}_U)$                     ▷ *Predict class probabilities for $\mathcal{D}_U$.*
11:      $\mathcal{D}_{L_{self}}^{conf}, \mathcal{D}_U^{unconf}, \mathcal{D}_U^{rest} \leftarrow \text{ASSIGNSETS}(\mathcal{D}_U, \hat{Y}, \mathbf{z}, \sigma, \rho, c)$
12:      $DS \leftarrow \mathcal{D}_U^{rest}$
13:      $AS, OS \leftarrow \text{CONSTRUCTSETS}(\mathcal{D}_{L_{self}}^{conf}, \mathcal{D}_U^{unconf}, s, o, r)$
14:      **if** $AS = \emptyset$ **then return** $\mathcal{D}_L$    ▷ *MetaEnsemble $\mathcal{E}$ is uncertain $\rightarrow$ abstain from self-labeling.*
15:      **end if**
16:      $\mathcal{D}_L \leftarrow \mathcal{D}_L \cup AS \cup OS$                 ▷ *Update labeled dataset $\mathcal{D}_L$.*
17:      $\mathcal{D}_U \leftarrow \mathcal{D}_U \setminus \{DS \cup AS \cup OS\}$          ▷ *Update unlabeled dataset $\mathcal{D}_U$.*
18:      $iter \leftarrow iter + 1$
19: **end while**
20: **return** $\mathcal{D}_L$

---

[t] Algorithm 3 constructs $\mathcal{D}_{L_{self}}^{conf}$ from all instances, where all algorithms agree on and TabPFN is confident enough. All instances, where the MetaEnsemble is maximally uncertain and TabPFN as well, belong to $\mathcal{D}_U^{unconf}$. The rest, i.e., the instances where a majority of the algorithms inside MetaEnsemble disagrees with a minority, belongs to $\mathcal{D}_U^{rest}$.

---

**Algorithm 3** AssignSets

---

**Require:** Unlabeled dataset $\mathcal{D}_U$, Predicted hard labels $\hat{Y}$, Predicted class probabilities $\mathbf{z}$, Confidence threshold $\sigma$, Uncertainty threshold $\rho$, Number of algorithms inside MetaEnsemble $c$

  1: $\mathcal{D}_U^{conf} \leftarrow \{\mathbf{x}_i \in \mathcal{D}_U \mid h_1(\mathbf{x}_i) = \ldots = h_c(\mathbf{x}_i), z_1 \notin [1 - \sigma, \sigma]\}$
  2: $\mathcal{D}_{L_{self}}^{conf} \leftarrow \{(\mathbf{x}_i, h_1(\mathbf{x}_i)) | \mathbf{x}_i \in \mathcal{D}_U^{conf}\}$          ▷ *Construct confident dataset.*
  3: $\mathcal{D}_U^{unconf} \leftarrow \{\mathbf{x}_i \in \mathcal{D}_U \mid \sum_{j=1}^c h_j(\mathbf{x}_i) = \frac{c}{2}, z_1 \in (1 - \rho, \rho)\}$    ▷ *Construct uncertain dataset.*
  4: $\mathcal{D}_U^{rest} \leftarrow \{\mathbf{x}_i \in \mathcal{D}_U | \mathbf{x}_i \notin \{\mathcal{D}_U^{conf} \cup \mathcal{D}_U^{unconf}\}\}$
  5: **return** $\mathcal{D}_{L_{self}}^{conf}, \mathcal{D}_U^{unconf}, \mathcal{D}_U^{rest}$

---

Algorithm 4 computes, how many instances from class 0 and class 1 will be sampled for self-labeling, based on the class proportion $r$. The instances are sampled uniformly from $\mathcal{D}_{L_{self}}^{conf}$ to construct the agreement set $AS$. From $\mathcal{D}_{U}^{unconf}$ the instances are sampled uniformly to construct the oracle set $OS$. The instances inside $OS$ are then labeled by the oracle.

---

**Algorithm 4** ConstructSets

---

**Require:** Self-labeled, confident dataset $\mathcal{D}_{L_{self}}^{conf}$, Uncertain dataset $\mathcal{D}_{U}^{unconf}$, Number of instances to self-label $s$, Number of instances to forward to the oracle $o$, Class proportion positive class $r$

1: $AS \leftarrow \{(\mathbf{x}_i, h_1(\mathbf{x}_i)) \sim \mathcal{D}_{L_{self}}^{conf} \}$      $\triangleright$ *Sample first instance randomly.*
2: **if** $p \leq 0.5$ **then** $s_0 \leftarrow s - \lfloor r \cdot s \rfloor$      $\triangleright$ *Compute parameter for stratified sampling.*
3: **else** $s_0 \leftarrow \lfloor r \cdot s \rfloor$
4: **end if**
5: **while** $|AS| < s$ **do**
6:      $iter \leftarrow 0$
7:      **while** $iter < s_0$ **do**      $\triangleright$ $s_0$ *many instances with pseudo-label 0.*
8:          $AS \leftarrow AS \cup \{(\mathbf{x}_i, h_1(\mathbf{x}_i)) \sim \mathcal{D}_{L_{self}}^{conf} | h_1(\mathbf{x}_i) = 0\}$
9:          $iter \leftarrow iter + 1$
10:      **end while**      $\triangleright s - s_0$ *many instances with pseudo-label 1.*
11:      $AS \leftarrow AS \cup \{(\mathbf{x}_i, h_1(\mathbf{x}_i)) \sim \mathcal{D}_{L_{self}}^{conf} | h_1(\mathbf{x}_i) = 1\}$
12: **end while**
13: $OS_U \leftarrow \{x_i \sim \mathcal{D}_{U}^{unconf} \}$      $\triangleright$ *Sample first instance randomly.*
14: **while** $|OS_U| < o$ **do**
15:      $OS_U \leftarrow OS_U \cup \{\mathbf{x}_i \sim \mathcal{D}_{U}^{unconf}\}$
16: **end while**
17: $OS \leftarrow \{(\mathbf{x}_i, O(\mathbf{x}_i)) | \mathbf{x}_i \in OS_U\}$      $\triangleright$ *Forward instances in $OS_U$ to the oracle for labeling.*
18: **return** $AS, OS$

---

## C EXPERIMENTS

This section contains further information on the experiments.

### C.1 MODEL DESCRIPTIONS

**ML Models.** The following traditional (simple) ML models are used in our experiments.

- **Logistic Regressor**
  Logistic regression is a binary classification algorithm that models the probability of a binary outcome using a logistic (sigmoid) function. It is a variant of linear regression, which transforms the linear output into probabilities.

- **Random Forest**
  Random Forest is a powerful ensemble learning method. It works by constructing multiple trees and combining their predictions. Each tree is trained on a random subset of the data, and the final prediction is made by aggregating the outputs of these trees. This approach reduces overfitting and improves the generalization of the model.

- **KNN-Classifier**
  K-Nearest Neighbors is an instance-based classification algorithm. It classifies data points by finding their K-nearest neighbors in the training data and assigning the majority class label among those neighbors to the test point. K is a user-defined parameter that determines the number of neighbors to consider.

- **Support Vector Machine (SVM) with a Linear Kernel**
  The SVM is a binary classification algorithm that seeks to find a hyperplane that best separates the classes in the data. When a linear kernel is used, SVM aims to create a linear decision boundary. It does this by maximizing the margin between the two classes, which improves the robustness of the model.

- **SVM with a Radial Basis Function (RBF) Kernel**
  The SVM with an RBF kernel is an extension of SVM for handling nonlinearly separable data. It maps the data into a higher-dimensional space where a linear decision boundary can be established. The RBF kernel is particularly useful when the data doesn not have a clear linear separation.

- **Gaussian Process Classifier**
  Gaussian Process is a probabilistic classification model that provides not only predictions but also uncertainty estimates. It models the distribution over functions and can adapt to the data. This makes it valuable for small datasets or situations where uncertainty in predictions is crucial.

- **Decision Tree**
  A Decision Tree is a hierarchical model that makes decisions based on feature values. It recursively splits the data into subsets based on the most informative features, aiming to maximize information gain or minimize impurity at each step.

- **Adaboost Classifier**
  AdaBoost is an ensemble learning technique that combines multiple weak classifiers into a strong classifier. It assigns weights to data points, focusing more on those that are misclassified in each iteration. By iteratively improving the classification of difficult examples, AdaBoost creates a robust final classifier.

- **Gaussian Naive Bayes Classifier**
  The Gaussian Naive Bayes classifier is a probabilistic algorithm based on Bayes' theorem. It assumes that features follow a Gaussian distribution and calculates the probability of a data point belonging to a particular class.

**AutoML Tools.** The following AutoML-tools are used in our experiments as well as for the baselines.

- **AutoGluon**
AutoGluon Erickson et al. (2020), an open-source AutoML framework, employs a sophisticated ensemble technique known as multi-layer stacking to enhance predictive model performance. The method begins by training a set of base models. These base models generate predictions, which are then used as features in subsequent stacking layers. Each stacking layer consists of stacker models that learn from the base models predictions and the original dataset, creating a hierarchical structure. This approach captures diverse patterns, reduces overfitting, and leverages all available data for training. The final stacking layer aggregates predictions using ensemble selection, resulting in highly accurate model predictions. The multi-layer stacking strategy enhances predictive robustness and generalization, making it a powerful tool for constructing accurate predictive models for structured tabular data in a user-friendly and predictable manner.

- **TabPFN**
TabPFN Hollmann et al. (2023) is a cutting-edge model for fast and accurate supervised classification on small tabular datasets. It uses in-context learning (ICL) to make predictions based on labeled examples in the input, eliminating the need for further parameter updates. TabPFN is trained offline on synthetic data using a Prior-data Fitted Network (PFN) architecture, which incorporates a novel prior specifically designed for tabular data. During training, a 12-layer Transformer model is trained on synthetic datasets generated from the prior. This training step is done once and takes advantage of architectural enhancements for efficient inference. During inference, TabPFN approximates the posterior predictive distribution (PPD) for the dataset prior, favoring simpler and causal explanations for the data. TabPFN achieves exceptional speed and competes with complex AutoML systems on small tabular datasets, making it a valuable tool for rapid classification tasks.

**SSL Tools.** Additionally, we compared our approach to the following two baselines.

- **Snuba**
Snuba Varma & Ré (2018) is a system designed to automate the process of generating heuristics for assigning training labels to large, unlabeled datasets in a weak supervision setting. It aims to replace the human-driven heuristic development process, which can be time-consuming and expensive. Snuba leverages a small labeled dataset to automatically create noisy heuristics that iteratively label portions of the unlabeled data. Importantly, it introduces a statistical measure that ensures the termination of this iterative process before the heuristics degrade label quality beyond a certain threshold. The core components of Snuba include a synthesizer for accuracy, a pruner for diversity, and a verifier to determine termination conditions. These components work together to generate a set of heuristics that balance accuracy and coverage while maintaining label quality. Its system architecture and theoretical guarantees ensure its efficiency and effectiveness in automating the weak supervision process. However, this approach heavily relies on the feature representation of each instance, which has to be chosen by the user. The authors only give a rough idea of how these feature representations could be generated.

- **VIME**
VIME Yoon et al. (2020) (Value Imputation and Mask Estimation) is a self- and semi-supervised learning framework designed for tabular data. Unlike existing methods that excel in structured data like images and language, VIME adapts to general tabular data, which lacks the same explicit structure. It introduces two pretext tasks: feature vector estimation and mask vector estimation for self-supervised learning, along with a novel tabular data augmentation method. VIME creates augmented samples for each training sample by applying the mask vector estimation and feature vector estimation procedures multiple times. These augmented samples are used to calculate the unsupervised (consistency) loss during training. After training, the predictive model can make predictions on new test samples.