# OpenReview forum: "Active Automated Machine Learning with Self-Training"
_ICLR.cc/2024/Conference — Submitted to ICLR 2024_

### Official Review · Reviewer_HF5Z · 2023-10-26

**Soundness:** 3 good
**Presentation:** 3 good
**Contribution:** 3 good
**Rating:** 6
**Confidence:** 4

**Summary:**

This paper introduces an augmented AutoML system AutoActiveSelf-Labeling (AutoASL) for semi-supervised tabular data tasks. AutoASL combines traditional ML and AutoML algorithms and incorporates strategies from Self-training and Active Learning to leverage information from unlabeled data. The method has a certain rationality and feasibility. The paper has a clear structure and clear hierarchy.

**Strengths:**

This paper creative combinations of existing methods.

**Weaknesses:**

The contribution lacks novelty creative as the paper only combinations of existing methods.

**Questions:**

(1)	Should the line 2 and line 3 in Algorithm 1 be swapped?
(2)	In page 6, the formula of DS does not match the textual description.
(3)	For DS, what the “majority of the predictors” and the “minority” refer to is unclear.
(4)	In Table 1, out of 47 datasets, only 22 had the best results, which is less than half of the total number of results.

---

> ### Author Response · Authors · 2023-11-14
>
> Thank you for providing valuable feedback. We genuinely appreciate your insights and suggestions. In light of your comments, we recognize the need for improvement, and as a result, we have decided to withdraw our paper. We look forward to addressing the identified areas for improvement and aim to submit an improved version in the future. Thank you again for your constructive feedback.
>
> Regarding your questions:
> Q1: You are right, these lines should be switched.
> Q2: Also true.
> Q3: We explained this explicitly in the description of Figure 2. But maybe we should also emphasize it more in the main body.

---

### Official Review · Reviewer_VFs5 · 2023-10-28

**Soundness:** 2 fair
**Presentation:** 2 fair
**Contribution:** 2 fair
**Rating:** 3
**Confidence:** 3

**Summary:**

**The paper presents AutoActiveSelf-Labeling (AutoASL), a method aiming to bridge the gap in Automated Machine Learning (AutoML) by utilizing unlabeled data by synthesizing self-training and active learning principles.** While the initiative to amalgamate these domains is commendable, the execution and substantiation within the paper leave room for substantial improvement and clarification.

A salient issue emerges in portraying the method as a symbiosis of self-training and active learning. **The integration of active learning, necessitating access to an oracle for true labels, deviates from the inherent autonomy of self-training.** This intersection raises conceptual ambiguities, as the reliance on an oracle for true labels in active learning seems to compromise the self-reliance and intrinsic labeling mechanism characteristic of self-training. Such a blend appears to muddy the clear delineation traditionally maintained between these two methodologies, necessitating a more explicit justification or clarification regarding the algorithm’s classification and operational principles.

Another notable shortcoming lies in the scope of experiments. The authors primarily tailor the AutoASL system to binary classification tasks, which curtails its generalizability and applicability across a broader spectrum of real-world problems. The omission of multi-class settings from the study's purview signifies a missed opportunity to showcase the method's versatility and robustness.

In conclusion, while the paper does introduce a unique perspective within the AutoML domain, it tends to be circumscribed by limited applicability, a lack of rigorous theoretical foundation, and insufficient engagement with contemporary methodologies and advancements. These aspects warrant critical reflection and enhancement to bolster the method's credibility, robustness, and relevance within the evolving landscape of AutoML.

**Strengths:**

- The paper takes a methodological step forward by attempting to unify self-training and active learning within a semi-supervised learning framework. This combination seeks to utilize both labeled and unlabeled data more effectively, marking an approach to enhancing existing systems.

- Regarding practical application, the proposed AutoASL system has demonstrated promising results in benchmark tests. This indicates a level of competence and potential usefulness in automated machine learning (AutoML), contributing to the ongoing advancements in the field.

**Weaknesses:**

- **Conceptual Clarity:** The paper presents a method as a fusion of self-training and active learning, but there seems to be a conceptual misalignment in this integration. The inclusion of an oracle for true labels in the active learning component seems at odds with the autonomous nature of self-training. This amalgamation raises questions regarding the true autonomy of the proposed method and muddles the distinct identities traditionally associated with each methodology. A clearer justification or elaboration on this aspect would be beneficial for understanding the algorithm’s unique operational principles.


- **Experimental Scope:** The current study predominantly focuses on binary classification tasks, limiting the algorithm's demonstrated applicability and generalizability. The absence of experiments involving multi-class settings restricts insights into the versatility and robustness of the proposed AutoASL system. Expanding the range of experiments to encompass diverse classification scenarios would have enriched the evaluation and illustrated the algorithm's adaptability to varied real-world challenges.

- **Hyperparameter Sensitivity:** The method exhibits a pronounced sensitivity to hyperparameters, which poses practical usability challenges. If users cannot ascertain and set hyperparameters effectively a priori, it could hinder the practical deployment and user-friendliness of this framework, making this aspect a significant limitation that needs addressing to bolster the model's utility in real-world applications.

- **Reference to Contemporary Works:** There’s a noticeable lack of engagement with recent advancements and scholarly works in both self-training and active leanring. Incorporating updated references and literature is essential to enhance the paper's contextual depth, comparative analysis, and alignment with contemporary scholarly discourse.

**Questions:**

- Could the authors elucidate the role of the oracle within the active learning phase of the algorithm? Specifically, is there a reliance on true labels during this step, and if so, to what extent? Understanding the degree of labeling required by the oracle is crucial for a comprehensive evaluation of the model's autonomy and practical applicability in semi-supervised learning contexts.

- What guidance or strategy does the paper offer regarding the a priori selection of hyperparameters? Clear instructions or criteria for choosing hyperparameters beforehand are crucial for users who aim to apply this framework effectively in practical scenarios. Your elucidation on this aspect would significantly contribute to the model’s usability and overall applicability.

- Could the authors provide an ablation study for AutoASL? It would be insightful to discern which specific components contribute most significantly to the overall performance of the system.

- The authors suggest extending the model to multi-class classification is relatively straightforward. Given this assertion, could you elucidate which components—between self-training and active learning—will likely be more pivotal or influential in a multi-class scenario? A clear delineation of the impact or contribution of each component in multi-class settings would be instrumental in understanding and applying the model effectively in broader contexts.

---

> ### Author Response · Authors · 2023-11-14
>
> Thank you for providing such valuable and extensive feedback. We genuinely appreciate your insights and suggestions. In light of your comments, we recognize the need for improvement, and as a result, we have decided to withdraw our paper. We look forward to addressing the identified areas for improvement and aim to submit an improved version in the future. Thank you again for your constructive feedback.
>
> Regarding your questions:
> Q1+3: You are completely right. While doing ablation studies where we
> first incorporated only the self-labeling component and then only the active learning component, we actually found out, that the active learning component plays a major role. Therefore we definitely need more understanding mabe also about differences regarding binary or multiclass settings.
>
> Q2: That is true, the choice of the hyperparameters is crucial and would be, in the best case, not hard coded, but chosen by the framework during inference (depending on the data distribution or whatsoever).
>
> Q3: Still to be found out, interesting point!

---

### Official Review · Reviewer_pCRY · 2023-10-31

**Soundness:** 1 poor
**Presentation:** 3 good
**Contribution:** 2 fair
**Rating:** 3
**Confidence:** 3

**Summary:**

The paper proposes AutoML system AutoActiveSelf-Labeling (AutoASL), which combines semi-supervised learning and active learning to AutoML. SSL is applied using pseudo-labeling techniques, AL is applied through an ensemble-based uncertainty measure. The proposed system is evaluated against state-of-the-art baselines across a diverse set of real-world datasets.

**Strengths:**

The paper is well-motivated to combine SSL and AL to AutoML as conventional AutoML methods only focused on supervised learning.

The proposed system is thoroughly evaluated against state-of-the-art baselines across a diverse set of real-world datasets.

**Weaknesses:**

Based on my understanding of the experiment description, the main weakness is the comparison lacks fairness. The baseline methods are only provided with 25 labeled instances, while the proposed AutoASL benefits from extra labeled instances obtained from its active learning procedure. For a fairer comparison, the baseline methods should also be provided with a comparable number of randomly selected samples.

In page 6, the equation of DS is not correct. By definition, the DS should also exclude instances for which the probability p(x) falls outside the range of (rho, 1-rho).

**Questions:**

Why disagreement set should be removed from the unlabeled set? I think labeling them by oracle should also be beneficial? The paper claims that removing these items "prevents them from getting wrongly labeled in future iterations", then why items in unconfident set are kept?

Why in Algorithm 1, line 5, if no models have sufficiently high accuracy score, the algorithm should stop and get classifier from labeled dataset by TabPFN? I think collecting true labels with AL should still be beneficial to improve the classifier .

---

> ### Author Response · Authors · 2023-11-14
>
> Thank you for providing valuable feedback. We genuinely appreciate your insights and suggestions. In light of your comments, we recognize the need for improvement, and as a result, we have decided to withdraw our paper. We look forward to addressing the identified areas for improvement and aim to submit an improved version in the future. Thank you again for your constructive feedback.
>
> Regarding your questions:
> Q1: Yes, of course labeling by oracle could be beneficial. But since we wanted to use the oracle as rare as possible, thats why we only labeled a subset of the instances inside the oracle set. (and discarding the disagreement set and the remaining instances in the oracle set)
>
> Q2: During our experiments, we found out, that in those cases, where none of the algorithms has a sufficiently high accuracy score (i.e.
> could learn the data distribution), this means, that the data is just really hard to learn/noisy etc. And even having more labeled data would not help in improving the accuracy of the classifiers.

---

### Official Review · Reviewer_XfBj · 2023-10-31

**Soundness:** 2 fair
**Presentation:** 3 good
**Contribution:** 2 fair
**Rating:** 3
**Confidence:** 4

**Summary:**

This paper presents AutoASL, an augmented AutoML system that combines principles from self-training and active learning to effectively leverage unlabeled data. Experimental results on open-source datasets show an improvement in the overall accuracy compared to state-of-the-art tools.

**Strengths:**

1. The problem this paper tackles is important. This paper explores the synergies between AutoML, SSL, and AL and demonstrates how methods from AutoML can effectively address SSL tasks.

2. This paper is clear and generally well-written.

**Weaknesses:**

1. The combination of SSL and AL is not new, and the author further introduced AutoML to the combination process. However, the motivation for introducing AutoML is unclear. As shown in Figure 1, it is just used for building the ensemble process, making the method more complex.

2. As we know, the AutoML process is time-consuming, so it is better to compare the training time of AutoASL and other tools.

3. The experimental results only show the average accuracy, which is not enough. For example, this paper mentions multiple parameters, but it does not provide a method for determining the optimal values of these parameters, nor does it conduct sensitivity analysis. Besides, since this system is a combination of many techniques, it is better to add an ablation study part to measure the influence of each part.

**Questions:**

See the Weaknesses

---

> ### Author Response · Authors · 2023-11-14
>
> Thank you for providing valuable feedback. We genuinely appreciate your insights and suggestions. In light of your comments, we recognize the need for improvement, and as a result, we have decided to withdraw our paper. We look forward to addressing the identified areas for improvement and aim to submit an improved version in the future. Thank you again for your constructive feedback.

---

### Meta-Review · Area_Chair_KpXh · 2023-11-27

**Metareview:**

After reviewing the feedback from reviewers, the authors have decided to withdraw their submission and submit an improved version in the future.

**Justification For Why Not Higher Score:**

The authors have indicated they would like to withdraw the submission.

**Justification For Why Not Lower Score:**

The authors have indicated they would like to withdraw the submission.

---

### Decision · Program_Chairs · 2024-01-16

Reject